# Hematopoiesis and Mast Cell Development

**DOI:** 10.3390/ijms241310679

**Published:** 2023-06-26

**Authors:** Domenico Ribatti, Antonio d’Amati

**Affiliations:** Department of Translational Biomedicine and Neuroscience, School of Medicine, University of Bari “Aldo Moro”, 70124 Bari, Italy; antonio.damati@uniba.it

**Keywords:** bone marrow, hematopoiesis, liver, mast cell, yolk sac

## Abstract

Hematopoietic stem cells (HSCs) are defined based on their capacity to replenish themselves (self-renewal) and give rise to all mature hematopoietic cell types (multi-lineage differentiation) over their lifetime. HSCs are mainly distributed in the bone marrow during adult life, harboring HSC populations and a hierarchy of different kinds of cells contributing to the “niche” that supports HSC regulation, myelopoiesis, and lymphopoiesis. In addition, HSC-like progenitors, innate immune cell precursors such as macrophages, mast cells, natural killer cells, innate lymphoid cells, and megakaryocytes and erythrocyte progenitor cells are connected by a series of complex ontogenic relationships. The first source of mast cells is the extraembryonic yolk sac, on embryonic day 7. Mast cell progenitors circulate and enter peripheral tissues where they complete their differentiation. Embryonic mast cell populations are gradually replaced by definitive stem cell-derived progenitor cells. Thereafter, mast cells originate from the bone marrow, developing from the hematopoietic stem cells via multipotent progenitors, common myeloid progenitors, and granulocyte/monocyte progenitors. In this review article, we summarize the knowledge on mast cell sources, particularly focusing on the complex and multifaceted mechanisms intervening between the hematopoietic process and the development of mast cells.

## 1. Introduction

Hematopoietic stem cells (HSCs) are defined based on the capacity to replenish themselves (self-renewal) and give rise to all mature hematopoietic cell types (multi-lineage differentiation) over the lifetime of an animal. As HSCs differentiate, they give rise to progenitors, which lack self-undergo lineage commitment to one of the renewal capacities, but in distinct blood lineages [1]. Isolation of murine HSCs predated the isolation of human HSCs. Starting in the 1980s, many cell surface markers have been identified that enrich for HSC activity in the mouse. However, no combination has been able to isolate pure HSCs. CD34 was the earliest marker for human HSCs discovered and is still of great use today. It took several years before additional markers that further define the human HSPC population were discovered. Perhaps it was due to difficulties in obtaining human material, or the primary research interest was more focused on the murine system. Studying human HSCs is quite difficult compared to murine. There is just as much scientific effort devoted to understanding how human HSCs are regulated and function, as to developing new technologies, assays, and enrichment and isolation strategies for those same purposes. 

Blood cells have been traditionally categorized into two basic branches: lymphoid and myeloid. The lymphoid lineage consists of T, B, and natural killer (NK) cells, which carry out adaptive and innate immune responses. The myeloid lineage includes several morphologically and functionally distinct cell types, including granulocytes (neutrophils, eosinophils, mast cells, and basophils), monocytes, erythrocytes, and megakaryocytes. 

During early human development, the hematopoietic process is divided into three stages. It begins in the yolk sac, and as the embryo and fetus develop, the fetal liver becomes the major hematopoietic organ, with the spleen also involved. During late fetal development and after birth, the bone marrow progressively takes over and becomes the main site in which blood cells form. 

The first site of hematopoiesis is the extraembryonic yolk sac, where mesodermal cells aggregate into clusters to form blood islands. Human yolk sac hematopoiesis begins at three weeks of gestation and consists predominantly of large nucleated primitive erythrocytes and primitive macrophages [2]. The blood islands are found within the yolk region adjacent to the allantois. Yolk sac blood islands are constituted of vascular endothelial cells and hematopoietic progenitor cells which give rise to primitive erythroblasts and primitive macrophages. At four postconceptional weeks, the yolk sac cell revealed the presence of HSCs-like progenitors, innate immune cell precursors such as macrophages, mast cells, natural killer cells, and innate lymphoid cells, as well as megakaryocytes and erythrocyte progenitors. Subsequently, monocyte and dendritic cell precursors and lymphocyte precursors appear in the fetal liver and definitive HSCs can form erythroid, megakaryocytic, myeloid, and lymphoid lineages [3]. The fetal liver is seeded with HSCs from the yolk sac around day 23, but there is also a second wave of hematopoiesis that originates from the wall of the newly formed aorta around day 27 and seeds the liver around day 30 where the HSCs expand at least two orders of magnitude. The fetal liver microenvironment not only supports the expansion of HSCs but also the differentiation of some of the HSCs into progenitors, making the fetal liver the primary site of hematopoietic expansion in the embryo. In postnatal hematopoiesis, HSCs, which are mainly distributed in the bone marrow, differentiate into blood cells of various lineages [4]. 

The most important principle of a model of hematopoiesis that emerged from early studies is the clonal origin of all hematopoietic cells from an HSC. In the early model of the hierarchy, mature hematopoietic cells were clonally derived from colony forming units (CFUs) [5]. CFUs are derived from a restricted progenitor of myeloid and erythroid lineages and are distinct from HSCs. HSCs are uniquely defined by their capacity to self-renew, or generate daughter stem cells, and differentiate into all hematopoietic cell types. In other words, cell division of stem cells is linked with either maintenance or loss of the stem cell state. By contrast, cell division of differentiated cells is linked with the loss of their developmental state and further differentiation. The only exception to this rule is the memory B and T cells, which proliferate clonally in response to specific antigens [6].

## 2. Yolk Sac and Liver Mast Cells

Initially, mast cells were proposed to derive from blood basophils or from the local histiocytic progenitor. In 1909, Lombardo reported that in human embryos mast cells first appeared in the eighth week of gestation [7]. Other cells have been reported as precursors of mast cells including mesenchymal cells [8], fibroblasts [9], pericytes [10], endothelial cells [11], and thymocytes [12]. The first source of mast cells is the extraembryonic yolk sac, where the first hematopoietic on embryonic day 7 (ED7) [13]. Primitive hematopoiesis is followed by the production of erythro-myeloid progenitors (EMPs) [2,14], which are generated between E8.5 and E10.5 from hemogenic endothelial cells of the yolk sac (Figure 1) [15,16]. The multipotent hematopoietic progenitors EMPs found in the yolk sac are a scarce population and have been characterized by the phenotypic markers: CD45 low c-kit/high AA4.1^+^. Due to the low number of these progenitors found within the yolk sac, studies were conducted to test the multipotency of these cells in vivo. These cells required a prior clonal expansion in vitro but they could give rise to all blood cell lineages in vitro and in vivo. The endothelial microenvironment in the tissue plays an important role in supporting the emergence of hematopoietic progenitors. By E9 EMPs differentiate in erythrocytes, granulocytes, megakaryocytes, macrophages, monocytes, and mast cells [15,17,18,19,20]. Mast cell progenitors are detected in the yolk sac at ED 9.5 by means of a limiting dilution assay and enter peripheral tissues where they complete their differentiation [18,19,21].

The intra-embryonic aorta-gonad-mesonephros (AGM) region generates fetal-restricted hematopoietic stem cells between E9.0 and E10.0 and adult-type hematopoietic stem cells from E10.5 onwards [19]. The AGM-derived HSCs were functional and could repopulate a lethally irradiated mouse. Explant cultures and repopulation studies in mice showed long-term multilineage reconstitution potential of only the intraembryonic para-aortic splanchnopleura (PSp)-AGM tissue and not of hematopoietic precursors found in the yolk sac prior to the onset of circulation.

Around ED 11.5, the fetal liver become the primary site of hematopoiesis (Figure 1). The pattern of hematopoiesis is different in the liver than that of the bone marrow. The liver is particularly erythropoietic whilst the bone marrow is particularly granulopoietic. The pre-hematopoietic fetal liver has been shown to be capable of supporting blood cell production if it was supplied with an exogenous source of hematopoietic stem cells. However, the pre-hematopoietic fetal liver did not become a site of active blood cell production when grafted into an irradiated adult host Mast cells reach a peak in the fetal liver on E15 [22], and a transient wave of Kit^+^ thy1 mast cell progenitors is detectable in fetal blood on E14.5, peaking at E15.5 [23]. 

Starting in late gestation, embryonic mast cell populations are gradually replaced by definitive, HSC-derived progenitors and yolk sac-derived mast cells largely disappear in most tissues by early adulthood. The skin is first seeded by mast cell progenitors around E15, and mature skin mast cells appear two days later. In the skin, mast cells of yolk sac origin are already reduced to half of the total mast cell population at birth and are lost completely by postnatal week 6 [19,20]. Adult mice maintain populations of committed mast cell progenitors in the bone marrow and spleen [24]. Perturbation of the maternal environment leads to the production of cytokines that cause epigenetic alterations in erythromyeloid progenitors and thereby the chronic activation of erythromyeloid progenitors-derived immune cells, including mast cells, after birth [25]. Around ED 16.5, the bone marrow begins to take over as the key site of hematopoiesis, which persists until after birth and into adulthood [21].

## 3. Bone Marrow Hematopoiesis

Bone marrow is the predominant hematopoietic tissue during adult life, harboring HSCs and a hierarchy of different hematopoietic cells. The hierarchy includes HSCs at the base and their derivatives such as multipotent progenitors, committed progenitors, and differentiated cells. Blood contains more than ten different mature cell types including erythrocytes, megakaryocytes/platelets, myeloid cells (monocyte/macrophages and granulocytes), mast cells, T- and B-lymphocytes, natural killer cells, and dendritic cells. 

In mice, hematopoietic cells are classified based on the absence of markers associated with blood cell lineage commitment (Lineage negative, Lin^−^). This population is further enriched for progenitors based on the expression of stem cell antigen-1 (Sca-1) and stem cell factor (SCF) receptor (c-Kit). 

The bone marrow is a complex microenvironment composed of different kinds of cells contributing to the “niche” that supports HSC regulation, myelopoiesis, and lymphopoiesis [26,27,28]. The concept of the stem cell “niche” was first postulated in 1978, when was proposed that niches have a defined anatomical location and, most importantly, the removal of stem cells from these niches results in their differentiation. In line with this hypothesis, after each division, HSCs remain in this anatomical and well-defined niche, while progenitor cells are spatially dislodged and differentiated. In adults, 75% of HSCs are quiescent (G0), and 8% enter the DNA synthesis (S) and self-renew or differentiate into progenitor cells [29,30]. In adulthood, HSCs reside in the bone marrow, where their niche is located. Structurally, the bone marrow microenvironment is a highly vascularized tissue where arteries, distal arterioles, and sinusoids contribute to cell ingress and egress and to the delivery of oxygen, nutrients, and growth factors. In this context, the role of vascular endothelial cells in HSC maintenance has been extensively studied.

One niche is the endosteal niche, including the bone and different cell types responsible for hematopoiesis. The endosteal niche is localized in the internal bone shell surface, close to the endocortical and trabecular surfaces. The endosteum includes bone-forming osteoblasts, which influence HSC maturation, bone-resorbing osteoclasts, and other cells including fibroblasts, macrophages, endothelial cells, and adipocytes are localized near the endosteum. The osteoblasts were the first niche cells shown to influence HSPC behavior, and initially, it was believed that the quiescent, LT-HSCs reside in their immediate vicinity at the surface of the endosteum [26].

The vascular niche includes different nursing cells regulating the hematopoiesis in direct crosstalk with the megakaryocytes, macrophages, and endothelial cells, acting as gatekeepers for the immune cells [31]. In the vascular niche endothelial cells, pericytes, and smooth muscle cells contribute to form blood vessels and release molecules involved in the recruitment of HSCs, endothelial progenitor cells (EPCs) and mesenchymal stem cells (MSCs) [32]. In this context, the vascular niche regulates stem cell mobilization and proliferation by releasing angiocrine factors, including vascular endothelial growth factor (VEGF), fibroblast growth factor-2 (FGF2), interleukins (IL)-1, 3, 6, transforming growth factor beta (TGFβ), platelet-derived growth factor beta (PDGFβ), endothelin, granulocyte-colony stimulation factor (G-CSF), granulocyte-macrophage-CSF (GM-CSF), and nitric oxide (NO) [33]. The closeness between sinusoidal endothelial cells and HSCs is crucial for their maturation [34]. FGF2 mediates the crosstalk between the vascular and the endosteal niche [35] and gives rise to a gradient between the endosteal and vascular niches and is involved in the recruitment of HSCs and their progenitors to the vascular niche [36].

Because of the limited life span of circulating hematopoietic cells, some HSCs are active and contribute to the replace old hematopoietic cells with new ones. Such a demanding job may lead to the exhaustion of HSCs; however, the balance between the quiescent and proliferative states of HSCs is tightly regulated by intrinsic and extrinsic factors of the surrounding niche. The HSC niche is composed of endothelial cells, osteoblasts, mesenchymal cells, and reticular cells. Among these components, endothelial cells and osteoblasts regulate HSC function. Using SLAM (CD150^+^, CD48, and CD244^−^) markers in identifying HSCs showed that 60% of these cells are located close to the sinusoidal endothelium, while 14% are in the endosteal zone [27,37]. It has been suggested that HSCs are in hypoxic zones close to endosteal areas where they are maintained in a quiescent state to avoid their exhaustion and differentiation [38,39,40,41], a characteristic that is crucial for their long-term repopulating activity [29,30]. However, more committed progenitors and cycling HSCs localize near vasculature in the bone marrow [27,32,42,43]. Time-lapse observation of the bone marrow with multi-photon microscopy demonstrated that transplanted HSCs detach from the endothelium and penetrate deep into the bone marrow, which is a hypoxic zone [44].

## 4. Bone Marrow Mast Cells

Early bone marrow transplantation experiments established the hematopoietic origin of adult mast cell populations [45,46,47]. In 1977, Kitamura and co-workers demonstrated the reconstitution of mast cells in mast cell-deficient W/Wv (mice were used that carry hypomorphic alleles of Kit, such as Kit W/Wv and Kit Wsh/Wsh mice. This strain bears two mutated alleles at the W locus: the W mutation is a G to A point mutation at a splice donor site leading to exon skipping and production of a truncated c-Kit, that lacks the transmembrane domain and is not expressed on the cell surface. In these mice, mast cells are absent or reduced because they depend on Kit signaling for their growth and survival) by transfer of wild-type bone marrow, reconstituted mast cells provide evidence that mast cells are derived from precursors that reside in the bone marrow. In vitro studies confirmed the hematopoietic origin of mouse mast cells [48]. In mice, EMPs give rise to either the megakaryocyte-erythrocyte progenitor or the granulocyte-macrophage progenitor, which further differentiates in macrophage, eosinophil, neutrophil, or to basophil-mast cell progenitor, expressing Kit, Fcƴ RII/RIII, β7 integrin [49]. 

In human postnatal hematopoiesis, bone marrow is the main source of various blood cell lineages including mast cells. However, the developmental trajectory of mast cells in the bone marrow and the existence of lineage-committed mast cell progenitors has remained unclear. The first identification of mast cell progenitors was conducted by Rodewald et al., isolating lineage committed in mouse fetal blood [23]. Other studies reported committed mast cell progenitors in adult bone marrow [50,51], peripheral blood, and peritoneal cavity [52,53]. Mast cells originate from the bone marrow where they develop from the HSCs via multipotent progenitors, common myeloid progenitors and granulocyte/monocyte progenitors (Figure 1) [54]. Progenitors giving rise to both mast cells and basophils have been isolated from bone marrow within the granulocyte/monocyte progenitor cell fraction. Most of these progenitors are found among the FcɛRI^+^ granulocyte/monocyte progenitors. However, progenitors that give rise to mast cells and basophils are also found within the FcɛRI−granulocyte/monocyte progenitors. Human mast cells derive directly from pluripotent CD34^+^, Kit^+^, CD13^+^, and CD117^+^ stem cells detectable in the bone marrow, peripheral blood, and peripheral tissues [55]. 

c-Kit (CD117) is expressed on HSCs and is retained on mast cells throughout their development and differentiation. After exposure to SCF, HSCs undergo maturation and develop into mature mast cells. c-Kit is the tyrosine kinase receptor for SCF [56]. In the adult, c-Kit is expressed by HSCs and is important for the maintenance of normal hematopoiesis and bone marrow mesenchymal stem cells. c-Kit is downregulated during terminal differentiation, but, among hematopoietic cells, mast cells retain high levels of expression of c-Kit also at the final stage of differentiation and are strongly dependent on SCF for their development, survival, and function; c-Kit is the most important factor involved in mast cell differentiation and deprivation of SCF results in mast cell growth arrest and apoptosis [55,57]. Administration of human recombinant SCF resulted in the development of an increased number of mast cells [58]. However, it has been also demonstrated that mast cell progenitors from peripheral blood survive, mature, and proliferate without SCF and KIT signaling in vitro [59].

Throughout life, bone marrow-derived progenitors further complement mast cells at mucosal sites and in conditions of environmental perturbations. Mast cell-committed progenitors express integrin α4β7, Cx3cr1, and c-Kit and undergo terminal maturation following recruitment into tissues. Mice lacking mast cells display peripheral neutrophilia and expansion of bone marrow HSCs [60], and hematopoietic reconstitution was accelerated in wild-type mice that received mast cell-deficient bone marrow and in mast cell-deficient recipients that received wild-type bone marrow [21]. Mast cell precursors circulate as agranular cells, traverse the vascular space, and enter the tissues, where they complete their development [23], originating specific subsets of mast cells under the influence of the local microenvironment [24]. The maturation of mast cells proceeds with the development of granules which evolve from vesicular pro-granules in the Golgi zone and increase in number and size. 

Several other cytokines are also important for mast cell development and/or maturation. In mice, the T cell-derived cytokine IL-3 is important for the proliferation of mouse mast cell populations from the spleen, intestine, and bone marrow. In comparison, mast cells that exhibit a more connective tissue-like phenotype in vitro require other factors such as SCF and/or IL-3 [61]. Although IL-10 by itself does not promote the proliferation of mast cells, it has synergistic effects with IL-3, SCF, or IL-4 in the proliferation of mouse mast cells. On the other hand, culturing human mast cells in vitro requires SCF and IL-6 [61]. In the presence of these cytokines, mast cell populations can be cultured from human cord blood-derived mononuclear cells, CD34^+^ HSCs, and peripheral blood mast cell precursors.

## 5. Transcriptomic Analysis

Ribonucleic acid (RNA) and messenger RNA (mRNA) are used by cells to function according to the information stored in their genomes. mRNA molecules encode the genetic information necessary to synthesize proteins and their abundance is, therefore, indicative of cell function. The field of transcriptomics is the scientific discipline that studies mRNA abundances of all genes in different cell types, tissues, species, and conditions (such as healthy/diseased, wild type/mutant, etc.). Until recently, microarrays and RNA sequencing were the dominant techniques in transcriptomics [62]. 

Studies performing transcriptional analysis collectively indicate an association between basophils, eosinophils, and mast cells, perhaps suggesting the existence of a common developmental precursor [63]. Saito et al. performed transcriptome analysis of mast cells derived from human umbilical cord blood and peripheral blood, revealing a series of mast cell-specific genes, e.g., TPSAB1 (tryptase α1 and β1), HDC (L-histidine decarboxylase), CTSG (cathepsin G), and CPA3 (carboxypeptidase A) [64]. Motakis et al. reported a comprehensive view of the transcriptome of mature skin mast cells in relation to other mature cell lineages, indicating limited relation between mast cells and other cell lineages including basophils [65]. Another study performed single-cell transcriptome analysis of primary bone marrow hematopoietic stem and progenitor cells to delineate the development path of multiple blood cell lineages, including one cluster expressing CLC and HDC, which are signature genes of basophil and/or mast cell lineages [66]. Moreover, analysis of flow cytometry data has allowed to the establishment a map of basophil and mast cell differentiation until the bifurcation of progenitors into two specific cell lines [67]. Wu et al. used single-cell transcriptomic analysis and demonstrated an association between the appearance of FcɛRI and the mast cell gene signature CD34^+^ hematopoietic progenitors in adult peripheral blood [68].

However, the lack of reliable signature genes for basophil and mast cell annotation makes the developmental trajectories indistinguishable. Functioning as the main granulocytes involved in type 2 inflammation, basophils, and mast cells show similar transcriptome profiles by expressing common marker genes, e.g., HDC, FcεRI encoding genes, and ENPP3. The low mRNA levels in mature human basophils make it difficult to characterize the basophils’ transcriptional profile in-depth. Further studies are needed to explore in-depth the transcriptional differences between human basophils and mast cells to better annotate and discriminate them.

## 6. Mast Cells in Different Pathological Conditions

Mast cells play multiple roles extending far beyond their classical role in Ig-E mediated allergic reactions, as it is summarized in Figure 2. For many years, mast cells have been considered involved in different allergic disorders including bronchial asthma, allergic rhinitis, urticaria, food allergy, atopic dermatitis, and angioedema. 

Mast cells change from protective immune cells to pro-inflammatory cells, capable of influencing the progression of several pathological conditions including mastocytosis, autoimmune diseases, neurodegenerative disorders, and tumors [69,70,71,72]. In cancer development, mast cells can promote tumorigenesis and invasion by releasing pro-inflammatory and angiogenic mediators such as TNFα, IL-6, VEGFs, and matrix-degrading enzymes. On the other hand, mast cell mediators, including TNF-α, IL-9, tryptase, histamine, granzyme B, and reactive oxygen specimens (ROS), have also been reported to mediate anti-cancer functions.

## 7. Concluding Remarks

Mast cells are tissue-resident multifunctional cells involved in various physiological and pathological processes. They were first discovered in the connective tissue by Paul Ehrlich in 1877 based on their special metachromatically staining characteristics. 

Mast cell heterogeneity is conferred by their ability to express certain profiles of proteases according to the environmental cues they encounter in their residential tissues. In humans, almost three different mast cell subpopulations derived from HSCs have been described based on their protease content: those that express tryptase (MC_T_), tryptase and chymase (MC_TC_), and chymase only (MC_C_), differing in their localization, mediator content, and responsiveness to secretagogues [73]. MC_T_ differ from their MC_TC_ counterpart in terms of anatomical location, staining characteristics, radiation sensitivity, and T cell dependence for development. MC_TC_ are predominantly located in the subepithelial layers of the skin within the peritoneal cavity and within the submucosal layers of the intestine, while MC_T_ are found in the lung and small intestine mucosa. It is important to note that patients with elevated tryptase levels should undergo further testing to find out the causative agent of a potential allergic reaction, and patients with normal tryptase concentration should undergo further diagnosis if they manifest clinical symptoms of a severe anaphylactic reaction. MC_TC_ have a long lifespan (probably several years) and have prominent granules that contain a large amount of histamine. Studies in mouse embryos found that MC_TC_ are mainly derived from late EMPs, while MC_C_ are mainly generated from HSCs. Otherwise, in humans, the origin of different mast cell subpopulations remains to be determined. More recent single-cell transcriptome-based analyses have confirmed the presence of heterogeneous mast cell subpopulations, but unlike the traditional discrete subdivision into two mast cell subsets, the transcriptional profiles of mast cells are continuous, thus dividing them into polarized and transitional subpopulations [74].

Mast cell-committed progenitors leave hematopoietic tissues, invade connective or mucosal tissue, proliferate, and differentiate into morphologically identifiable mast cells. Mature and differentiated mast cells are rarely found in the circulation of healthy individuals, but they have been found present in the bloodstream of patients suffering from systemic mastocytosis, a condition where mast cells have attained activating mutations in the c-Kit receptor that promote uncontrollable proliferation, activation, and degranulation of mast cells [69]. 

Mast cell progenitors appear in peripheral tissues before mature mast cells, and this event is an expression of the fact that fetal progenitors commit to the mast cell lineage prior to entering tissues and undergo further maturation there. After their differentiation and their establishment in different anatomical sites. Mast cells preferentially reside nearby surfaces exposed to environmental triggers such as skin, airways, and gastrointestinal tract. In these specific contexts, mast cells of different organs express peculiar factors, responsible for their tissue-specific functions [75]. These different locations pose mast cells in a particularly relevant position for the initiation and propagation of immune responses.

## Figures and Tables

**Figure 1 ijms-24-10679-f001:**
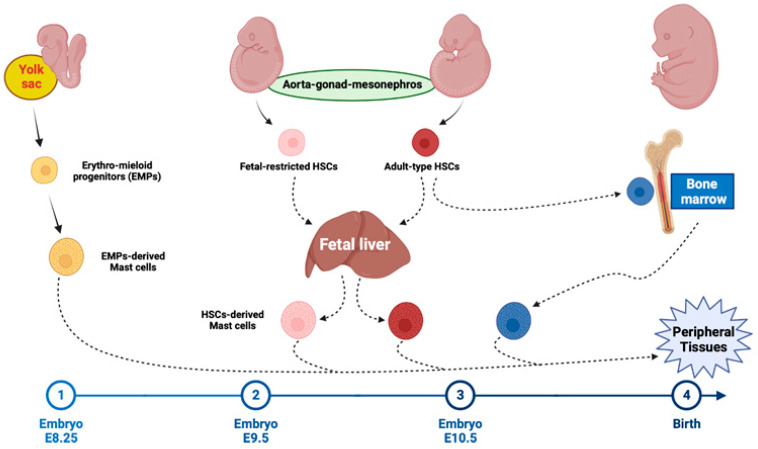
Mast cell development in the mouse. The first mast cell progenitors arise from the erythro-myeloid progenitors (EMPs) produced in the yolk sac starting at embryonic day 8.25 (E8.25). EMPs differentiate into mast cells via committed progenitors that may seed peripheral tissues directly from the yolk sac and/or through fetal liver intermediates. The intra-embryonic aorta-gonad-mesonephros (AGM) region generates fetal-restricted hematopoietic stem cells (HSCs) between E9.5 and E10.5. Both migrate to the fetal liver and may produce mast cells that supplement the first wave of EMP-derived mast cells. Around birth, adult-type HSCs colonize the bone marrow where they persist lifelong.

**Figure 2 ijms-24-10679-f002:**
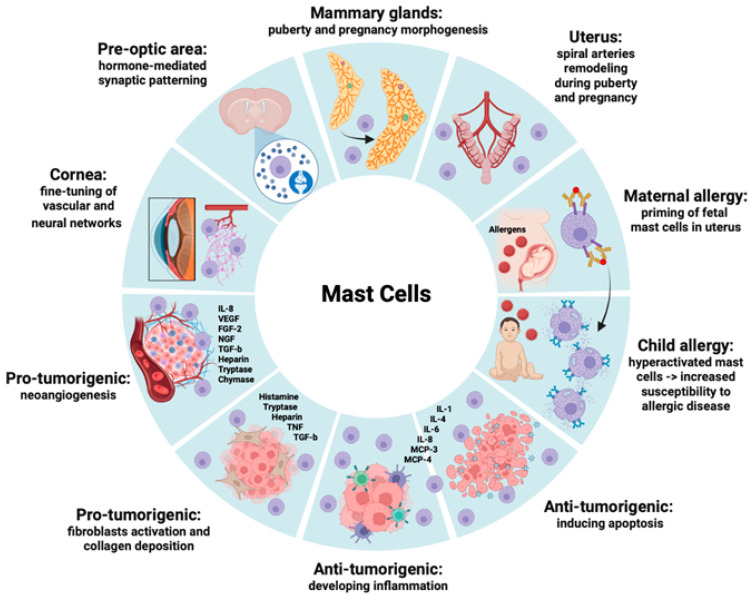
Mast cells and their mediator’s involvement in different physiological and pathological conditions.

## Data Availability

No new data were created or analyzed in this study. Data sharing is not applicable to this article.

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
