# Peer review of "Hematopoiesis and Mast Cell Development"

_ijms, 2023, doi:10.3390/ijms241310679_

Round 1

Reviewer 1 Report

The paper by Domenico Ribatti and Antonio d'Amati provides a comprehensive review of mast cells' origin, development and functionality. The article is well written, well structured and has precise figures illustrating significant parts of the text. The review summarises all critical information on mast cells and up-to-date research in the field. One crucial aspect is comparing and pointing out differences between mast cells and basophils used complementary in many clinical settings. There are only minor issues that could be dressed to increase paper discernability.
1.    Please consider adding short information on tryptase usage in diagnosing anaphylactic/anaphylactoid reactions.
2.    Please keep constant citing style – page 4, line 163; please add brackets for mentioned literature – page 6, line 251 (IL-480-82?) 

Author Response

There are only minor issues that could be dressed to increase paper discernability.

Please consider adding short information on tryptase usage in diagnosing anaphylactic/anaphylactoid reactions.

We have improved the MS as follows: “Patients with elevated tryptase levels should undergo further testing to find out the causative agent of a potential allergic reaction, and patients with normal tryptase concentration should undergo further diagnosis if they manifest clinical symptoms of a severe anaphylactic reaction”.

Please keep constant citing style – page 4, line 163; please add brackets for mentioned literature – page 6, line 251 (IL-480-82?) 

Done, we have corrected: IL-3 instead of IL-480-82 ?.

Reviewer 2 Report

Hematopoietic stem cells (HSCs) possess the ability to self-renew and generate various types of mature blood cells. HSCs primarily reside in the bone marrow, alongside other cells that contribute to the supportive environment for HSC regulation and blood cell production. The development of mast cells, as well as other innate immune cells and blood cell precursors, involves intricate relationships and transitions from embryonic to definitive stem cell-derived progenitors. This review provides an overview of mast cell sources and explores the complex mechanisms connecting hematopoiesis with mast cell development.

This is a very exciting and very well written review. Also, the topic is timely.

With the involvement of the niche; it will be useful to include about vascular niche, with the possibe impact with the lymphatics and blood vessels; and the changes with the age.... example - PMID:  36669473

Hematopoietic stem cells (HSCs) possess the ability to self-renew and generate various types of mature blood cells. HSCs primarily reside in the bone marrow, alongside other cells that contribute to the supportive environment for HSC regulation and blood cell production. The development of mast cells, as well as other innate immune cells and blood cell precursors, involves intricate relationships and transitions from embryonic to definitive stem cell-derived progenitors. This review provides an overview of mast cell sources and explores the complex mechanisms connecting hematopoiesis with mast cell development.

This is a very exciting and very well written review. Also, the topic is timely.

With the involvement of the niche; it will be useful to include about vascular niche, with the possibe impact with the lymphatics and blood vessels; and the changes with the age.... example - PMID:  36669473

Author Response

This is a very exciting and very well-written review. Also, the topic is timely. With the involvement of the niche, it will be useful to include about the vascular niche, with the possible impact on the lymphatics and blood vessels; and the changes with age.... example

We have improved the MS as follows:  “In the vascular niche endothelial cells, pericytes, and smooth muscle cells contribute to form blood vessels and release molecules involved in the recruitment of HSCs, endothelial progenitor cells (EPCs), and mesenchymal stem cells (MSCs ) (H.-G. Kopp, S.T. Avecilla, A.T. Hooper, S. Rafii, The bone marrow vascular niche: home of HSC differentiation and mobilization, Physiology 20 (2005) 349–356). In this context, the vascular niche regulates stem cell mobilization, and proliferation, by releasing angiocrine factors, including vascular endothelial growth factor (VEGF), fibroblast growth factor-2 (FGF2), interleukins (IL)-1, 3, 6, transforming growth factor beta (TGFβ), platelet-derived growth factor beta (PDGFβ), endothelin, granulocyte-colony stimulation factor (G-CSF), granulocyte-macrophage-CSF (GM-CSF), and nitric oxide (NO) (J. Pasquier, P. Ghiabi, L. Chouchane, K. Razzouk, S. Rafii, A. Rafii, Angiocrine endothelium: from physiology to cancer, J. Transl. Med. 18 (2020), 52-52). The closeness between sinusoidal endothelial cells and HSCs is crucial for their maturation (]D. Benayahu, U.D. Akavia, I. Shur, Differentiation of Bone marrow stroma-derived mesenchymal cells, Curr. Med. Chem. 14 (2007) 173–179). FGF2 mediates the crosstalk between the vascular and the endosteal niche (]G. de Haan, E. Weersing, B. Dontje, R. van Os, L.V. Bystrykh, E. Vellenga, G. Miller, In Vitro generation of long-term repopulating hematopoietic stem cells by fibroblast growth factor-1, Dev. Cell 4 (2003) 241–251), and gives rise to a gradient between the endosteal and vascular niches, and is involved in the recruitment of HSCs and their progenitors to the vascular niche (]D.E. Wright, Physiological migration of hematopoietic stem and progenitor cells, Science 294 (2001) 1933–1936).

Reviewer 3 Report

The authors present an interesting review article on the hematopoiesis of mast cells. 

MCTC and CTMC are only discussed in the conclusion but it is important to describe their differentiation from one another during development of mast cells from stem cells. 

The authors refer to MCTC as MMC and MCT interchangeably in the conclusion, they should stick to one and be consistent.

The authors should provide concrete examples of mast cell mediators described as having anti-cancer functions (line 335-337)

The authors should discuss where in the periphery mast cells are found.

The authors should discuss the longevity of mast cells.

Briefly the authors should discuss the role of mast cells in diseases. 

Myeloid is often spelled mieloid

line 251: what is IL480-82?

This review is novel and lacking in the field.

However, the authors do not go in depth and cover the topic in a superficial manner.

The authors spend half the paper discussing hematopoiesis at an introductory level. The bridge to mast cells and discussion of mast cells is very short. 

There are various typos and grammatical errors throughout the manuscript.

Figure 2 is the most interesting but is only discussed in the conclusion.

References are heavily lacking throughout the manuscript.

I recommend the authors re-work the manuscript and re-submit as a new submission.

Author Response

MCTC and CTMC are only discussed in the conclusion but it is important to describe their differentiation from one another during development of mast cells from stem cells. 

We have improved the MS as follows: “Mast cell heterogeneity is conferred by their ability to express certain profiles of proteases according to the environmental cues they encounter in their residential tissues. In humans, almost three different mast cell subpopulations derived from HSCs have been described based on their protease content: those that express tryptase (MCT), tryptase and chymase (MCTC), and chymase only (MCC), differing in their localization, mediator content, and responsiveness to secretagogues (Irani AA et al., 1986. Two types of human mast cells that have distinct neutral protease composition. Proc Natl Acad Sci USA, 83: 4464-4468). MCT differ from their MCTC counterpart in terms of anatomical location, staining characteristics, radiation sensitivity, and T cell dependence for development. MCTC are predominantly located in the subepithelial layers of the skin, within the peritoneal cavity, and within the submucosal layers of the intestine, while MCT are found in the lung and small intestine mucosa. MCTC have a long lifespan (probably several years) and have prominent granules that contain a large amount of histamine. Studies in mouse embryos found that MCTC are mainly derived from late EMPs, while MMC are mainly generated from HSCs. Otherwise, in humans the origin of different mast cell subpopulations remains to be determined.  More recent single-cell transcriptome-based analyses have confirmed the presence of heterogeneous mast cell subpopulations, but unlike the traditional discrete subdivision into two mast cell subsets, the transcriptional profiles of mast cells are continuous, thus dividing them into polarized and transitional subpopulations (Varricchi G et al., 2019. Future needs in mast cell biology. Int J Mol Sci, 20: 4397).

The authors refer to MCTC as MMC and MCT interchangeably in the conclusion, they should stick to one and be consistent.

We have deleted this statement.

The authors should provide concrete examples of mast cell mediators described as having anti-cancer functions (line 335-337)

We have improved the MS as follows: “On the other hand, mast cell mediators, including TNF-α, IL-9, tryptase, histamine, granzyme B, and reactive oxygen specimens (ROS), have also been reported to mediate anti-cancer functions.”

The authors should discuss where in the periphery mast cells are found.

We have improved the MS as follows: “After their differentiation and their establishment in different anatomical sites. Mast cells preferentially reside nearby surfaces exposed to environmental triggers, such as skin, airways, and gastrointestinal tract. In these specific contexts, mast cells of different organs express peculiar factors, responsible for their tissue-specific functions (Frossi B et al., 2018 . Is the time for new classification of mast cells? What do we know about mast cell heterogeneity. Immunol Rev, 282: 35-46).

The authors should discuss the longevity of mast cells.

We have improved the MS as follows. “MCTC have a long lifespan (probably several years) and have prominent granules that contain a large amount of histamine.” 

Briefly, the authors should discuss the role of mast cells in diseases. 

We have added a new paragraph as follows: “6. Mast cells in different pathological conditions. Mast cells play multiple roles extending far beyond their classical role In Ig-E mediated allergic reactions, as it is summarized in the Figure 2. For many years, mast cells have been considered involved in different allergic disorders, including bronchial asthma, allergic rhinitis, urticaria, food allergy, atopic dermatitis, and angioedema. Mast cells change from protective immune cells to pro-inflammatory cells, capable to influence the progression of several pathological conditions, including mastocytosis, autoimmune diseases, neurodegenerative disorders, and tumors (Volertas S et al., 2018. New insights into clonal mast cell disorders including mastocytosis. Immunol Allergy Clin North Am, 38: 341-350; Noto CN et al., 2021. Mast cells as important regulators in autoimmunity and cancer development. Front Cell Dev Biol, 9: 752350; Girolamo F et al., 2017. Immunoregulatory effect of mast cell influenced by microbes in neurodegenerative diseases. Brain Behav Immun, 65: 68-89; Ribatti D, 2013. Mast cells and macrophages exert beneficial and detrimental effects on tumor progression and angiogenesis- Immunol Lett, 152: 83-88). In cancer development, mast cells can promote tumorigenesis and invasion by releasing pro-inflammatory and angiogenic mediators such as TNFα, IL-6, VEGFs, and matrix-degrading enzymes. On the other hand, mast cell mediators, including TNF-α, IL-9, tryptase, histamine, granzyme B, and reactive oxygen specimens (ROS), have also been reported to mediate anti-cancer functions.

Myeloid is often spelled myeloid

We have corrected

line 251: what is IL480-82?

We have corrected: IL-3.

The authors spend half the paper discussing hematopoiesis at an introductory level. The bridge to mast cells and discussion of mast cells is very short. 

The MS is focused on the relationship between hematopoiesis and mast cell development.

There are various typos and grammatical errors throughout the manuscript.

We have corrected all the typos and grammatical errors.

Figure 2 is the most interesting but is only discussed in the conclusion.

We have moved the Figure 2 before.

References are heavily lacking throughout the manuscript.

We have improved the number of references.

Round 2

Reviewer 3 Report

At line 273, the authors wrote MMc when their abbreviation is MCc, they should fix this error.

Author Response

At line 273, the authors wrote MMc when their abbreviation is MCc, they should fix this error

We have corrected the error.